# An Overview of Managed Aquifer Recharge in Mexico and Its Legal Framework

**Mary Belle Cruz-Ayala [1],\* and Sharon B. Megdal [2]**

[1] The University of Arizona, 1064 E. Lowell St., Tucson, AZ 85721, USA

[2] Water Resources Research Center, The University of Arizona, Tucson, AZ 85721, USA; smegdal@email.arizona.edu

\* Correspondence: marybelca@email.arizona.edu

**Abstract:** In Mexico, one hundred of the 188 most important aquifers dedicated to agriculture and human consumption are over-exploited and 32 are affected by seawater intrusion in coastal areas. Considering that Mexico relies on groundwater, it is vital to develop a portfolio of alternatives to recover aquifers and examine policies and programs regarding reclaimed water and stormwater. Managed Aquifer Recharge (MAR) may be useful for increasing water availability and adapting to climate change in semi-arid regions of Mexico. In this paper, we present an overview of water recharge projects that have been conducted in Mexico in the last 50 years, their methods for recharge, water sources, geographical distribution, and the main results obtained in each project. We found three types of MAR efforts: (1) exploratory and suitability studies for MAR, (2) pilot projects, and (3) MAR facilities that currently operate. This study includes the examination of the legal framework for MAR to identify some challenges and opportunities that Mexican regulation contains in this regard. We find that beyond the technical issues that MAR projects normally address, the regulatory framework is a barrier to increasing MAR facilities in Mexico.

**Keywords:** MAR; Mexico; legal; regulatory; framework; LAN (Law of the Nation's Waters); reclaimed water; arid; semi-arid

## 1. Introduction

Groundwater overdraft in Mexico is a serious problem. One hundred of the 188 most important aquifer dedicated to agriculture and human consumption are overexploited, and 32 are affected by seawater intrusion in coastal areas [1,2]. Overexploitation of groundwater refers to the excessive withdrawal beyond the annual average recharge [1]. Expansion of irrigated agriculture, the growth in population, changes in consumption habits, and urbanization are the primary drivers of the increase in water use, resulting in water scarcity, a condition that is becoming a threat to sustainable development in Mexico [3]. Water scarcity refers to the geographic and temporal mismatch between freshwater demand and availability [3,4]. In Mexico, 58% of its national territory has dry lands—semi-arid, arid, and hyper-arid ecosystems [5]. Hence, natural conditions and overexploitation of groundwater [5] have led to a water-stress state in the country. Water stress refers to the ratio of total annual water withdrawals to the total available annual renewable supply [6].

Climate change is projected to affect rainfall patterns in Mexico; in particular, the Northern and Northwest regions may be facing severe water scarcity all year round by the year 2040. Mexican researchers analyzed weather databases to project rainfall patterns and temperature for the next four decades [7]. The information obtained was merged with data of water availability for overexploited aquifers; the outcomes showed that in the Northwest, Northern, and Northeast regions, water availability would decline [7,8]. For the US–Mexico border region, the results obtained by

US and Mexican scientists using several climate models, have shown "a continued high degree of annual precipitation variability" [9,10]. Regarding the temperature, average annual and seasonal temperatures, also nighttime temperature are projected to increase for the border US–Mexico region, the western Sonoran Desert, and the northern region of the Chihuahuan Desert [7]. The Working Group II Contribution to the IPCC Fifth Assessment Report pointed out that over North America, which includes Northwest Mexico, exhibit very likely increases in mean temperature [11].

Future climatic scenarios might also affect groundwater recharge because higher temperatures will increase potential evapotranspiration [9,12]. Groundwater recharge includes the process of recharge as a natural part of the hydrological cycle, human-induced recharge that is also named artificial recharge, and recharge due to human activities such as irrigation and waste disposal [13]. Considering that groundwater supplies 70% of the water used for industry, agriculture, and human consumption in Mexico and several aquifers are overexploited [1,14], alternatives must be evaluated to recover aquifers.

In Northwest Mexico, the states of Sonora and Baja California Sur are annually impacted by tropical storms that provide them with a large volume of water in a short period. The rainfall generates floods because there is no infrastructure to manage stormwater or store it. However, several cities in the same region lack water during periods of the year. Another example is Mexico City, which imports water from distant basins [15–18]. Nevertheless, this city's precipitation ranges between 500 and 1200 mm per year, and almost every summer it suffers floods [1,19]. Mexico City and its metropolitan area combine reclaimed water, raw water, and stormwater to discharge into streams without evaluating whether this water can be recycled for other activities [20]. Reusing water is particularly relevant in water-stressed countries like Mexico. Therefore, it is crucial to examine policies and programs regarding reclaimed water and stormwater, which might be useful resources for increasing water availability in this country.

Artificial recharge, as it is termed in Mexican law, now called Managed Aquifer Recharge (MAR), where water quality management is also explicitly addressed, is a concept applied to describe diverse methods with the aim of both augmenting groundwater resources during times when water is available and recovering the water from the same aquifer in the future when it is needed [21,22]. There are a number of systems that could be categorized as MAR [21,23,24] and to implement one of them depends on local conditions such as financial budget, water quality, land availability, hydrogeological characteristics of the aquifer, water availability, and type of soil, among other considerations [24–26]. Although MAR is not a cure for overexploited aquifers, it can be useful for restoring groundwater balance [21]. In Mexico, reclaimed water and stormwater are the primary sources for MAR. In several countries, Managed Aquifer Recharge using reclaimed water has been growing as an option to recycle this resource, to replenish aquifers, and to stop seawater intrusion [20,24,27–29]. Globally, metropolises that have successfully used MAR to increase water supply include Adelaide, Australia [30], Los Angeles, United States [21], Tucson, United States [31,32], and Orange County, United States [33].

*MAR in Mexico and Legal Framework*

Since the 1950s, researchers, consultants, and local governments have designed and implemented water recharge projects [25]. Even though some cases were not designed specifically for water recharge, it has been pointed out that these experiences revealed the potential of harvesting and storing water to be used in the future [34,35]. In the early 2000s, in the states of Chihuahua, Sonora, and Mexico City, facilities for MAR using reclaimed water were proposed.

The need to increase water supply in semi-arid lands has triggered efforts for MAR. For instance, in 2007, in San Luis Río Colorado, Sonora facilities for MAR began operating [36], although there were no regulations in place. Therefore, this case was used as a model to define national standards for MAR. Although academics have stressed that public policies for MAR are needed [37–39], Mexican federal agencies have made slow progress in this regard. Moreover, the role that national policies have played in promoting or impeding MAR applications in Mexico has not been analyzed.

In Mexico, the water administration is centralized in the federal government and its regulation is based on Article 27 of the Political Constitution of the United States of Mexico [40]. The municipal responsibilities for water are established in Article 115. Regarding MAR, the Law of the Nation's Waters establishes that a permit is required to build a MAR pilot project [41]. The Federal Duties Law includes the charges that users must pay to extract groundwater; this legislation contains financial incentives for artificial recharge, as well. Furthermore, there are two official Mexican standards for MAR [42,43]. These regulations establish guidelines and water quality parameters that MAR projects, using reclaimed water and stormwater, must meet.

Throughout the world, water policies have been grounded in hard-path solutions, creating large scale infrastructure to deliver water, capital-intensive supply sources and centralized management [44]. Mexico is no exception. Projects to manage water demand, recycle water, and build small-scale infrastructure for water recharge with decentralized management (soft-path solutions) have not often been incorporated in governmental water project portfolios. Particularly in water-stressed regions, hard and soft-path solutions can be simultaneously implemented to improve water management [44,45]. In Mexico, there are examples of small-scale facilities for MAR in rural communities and medium-sized cities that are helping to increase water supply. Hence, it is beneficial to evaluate the role that small infrastructure for MAR in Mexico could play to increase water availability in rural and peri-urban areas. In this paper, we are using the definition of water availability provided by Tydwell et al. [46]: water availability is the supply of water in excess of that currently allocated for consumptive use in a particular basin, that is, the amount of water available for new development.

In Mexico, researchers have conducted an inventory of projects and estimated a significant potential for MAR in Mexico [35,47]. This study provides additional and updated information, including operational status. We present an overview of water recharge projects that have been conducted in Mexico in the last 50 years, their methods for recharge, water sources, geographical distribution, and the main results obtained in each project. Previous research has been focused only on assessing the performance of MAR projects or suitability studies. Our study also includes the examination of the legal framework for MAR to identify some challenges and opportunities that the Mexican legislation contains in this regard.

## 2. Materials and Methods

We carried out a systematic literature review of peer-reviewed publications in English and Spanish language, using such keywords as MAR, Mexico, artificial recharge, and water recharge. In addition, considering that in Mexico several studies for MAR have been conducted by universities and private consultants, the search included grey literature and official reports. These reports are mainly the products that consultants have delivered to state and municipal authorities. Additionally, we obtained information from publicly available conference proceedings and academic theses and reviewed the Mexican regulatory framework for groundwater and wastewater.

Recharge projects were grouped based on whether reclaimed water or stormwater was the source of water. In this paper, we will refer to reclaimed water to describe water that has been treated to meet quality standards; the term effluent will be used for water that has not been treated. We also consider each project's objectives. Finally, we analyzed the Mexican framework regulating MAR to identify gaps or aspects that could be clarified. The Legislation analyzed included the Political Constitution of the United States of Mexico, the Law of the Nation´s Waters, Federal Duties Law, and Official Mexican Standards.

## 3. Results

### 3.1. MAR Projects, Types, Objectives, and Geographical Distribution

On the whole, we found that there are three types of MAR efforts (Table 1).

a. The first group includes exploratory and suitability studies for MAR, mainly proposed by researchers.

b. The second group is composed of pilot projects. In some cases, pilot projects have operated for a short period and later were stopped or canceled, and others are still operating. We use the definition of a pilot project for artificial recharge included in the Mexican regulation [40], where a pilot project is a temporary project that has been built to evaluate a recharge system, to assess its technical feasibility, to monitor hydraulic variables and water quality, and to identify possible impacts generated by artificial recharge in the aquifer and the environment.

c. The third group includes MAR facilities that currently operate.

Overall, MAR pilot projects and facilities have been planned with four objectives: to restore contaminated or depleted aquifers, to reduce land subsidence, to replenish aquifers and increase water availability, and to manage floods. Moreover, the infrastructure is mainly composed of small-scale facilities that are using stormwater or reclaimed water. In addition to the identified MAR projects and facilities, we include information on the most significant case of incidental recharge registered in Mexico, in the State of Hidalgo. Currently, only a portion of the water discharged meets quality standards; however, if this water is adequately treated, it could be used to recharge local aquifers. Finally, we present information regarding a MAR project that was created to recover the aquifer and to maintain environmental services in a riparian ecosystem. This information is presented and discussed in the sections below.

### 3.1.1. MAR Projects to Restore Contaminated or Depleted Aquifers

In a region between the states of Durango and Coahuila (Figure 1), a MAR pilot project was designed to find alternatives to halt arsenic contamination in aquifers used for human consumption and agriculture [48,49]. Using an infiltration reservoir that had previously been constructed in the basin, several experiments were conducted in 1991 and 2000. In those pilot projects, adequate rates of water infiltration and acceptable quality standards were obtained. However, this pilot project was not transformed into a large-scale installation because local farmers had provided most of the volume of water used for the project, and they were not interested in maintaining this experiment. In addition, the project lacked financial support for monitoring activities.

In Baja California Sur, Cardona et al. [50] analyzed the geological conditions in the municipality of Comondú to reduce seawater intrusion in its aquifers. Based on the information from Cardona [50], Wurl [51] evaluated three alternatives for recovering aquifers in this municipality. The first option is reducing 40 percent of withdrawals for agriculture; under this scenario, the aquifer would be recovered in more than 100 years [51]. The second option is to build small reservoirs to catch stormwater and slowly release it to increase natural recharge. The third option is the construction of storage dams for artificial recharge in areas with an adequate infiltration rate. Based on their hydrological models and maintaining the pumping rates registered in 2007, the last option would restore the aquifer at a faster pace. It would mean that in 40 years, 18% of the depleted volume of groundwater would have been restored [51,52].

### 3.1.2. MAR to Reduce Land Subsidence

Soil subsidence caused by depletion of groundwater has been widely documented in the United States, Mexico, and other countries [53–56]. In Mexico City, land subsidence has been recorded since 1925 and is one of the most notorious cases in the world [57,58]. Mexico City was built upon a lake over the ruins of the former Aztec city Tenochtitlan [59]. During the period 1991–2006 in some areas of this city, the subsidence rate has ranged 0.10–0.40 m per year [60,61]. Several studies argue that one of the leading causes of subsidence is that groundwater extraction exceeds the natural recharge and due to depressurization of the lacustrine aquitard [34,60–62]. It has been estimated that at least 23% of Mexico's City total surface area is placed on the former lakebed [57,59]. Therefore, to address the subsidence problem in Mexico City, some pilot projects using MAR methods have been proposed [34,63,64]. Since the 1960s, Figueroa Vega [62] has conducted studies and developed pilot projects for water recharge in Mexico City. Recently, Figueroa Vega [34] proposed a MAR project to collect water from

local streams and runoff during the rainy season. This proposal may be developed where an ancient lake was located, and it would be linked to the hydraulic facilities for water management that has been built in the metropolitan area of Mexico City [65]. Water collected could be used to recharge the aquifer and would help to reduce the subsidence rate.

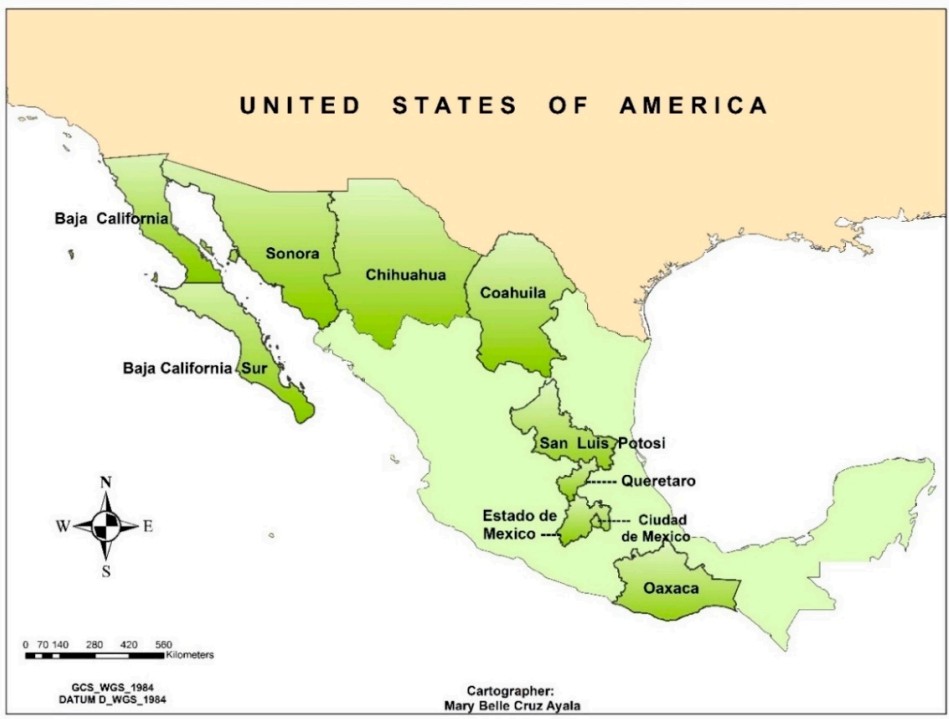

**Figure 1.** States in Mexico with the presence of Managed Aquifer Recharge (MAR) projects.

In Mexico City, another effect associated with over-pumping of groundwater and subsidence is the seismic damage in buildings and urban infrastructure. This city is a hot spot for earthquakes because it is situated between the North American Plate and the Cocos Plate [66]. Carreón-Freyre et al. [57] suggest that there is a relationship between seismic damage and land subsidence. Similarly, other scholars have developed a simulation model to identify urban areas located on top of the ancient lake and found that the most energetic shock waves during an earthquake are recorded in these zones [67]. Although more research is required to characterize specific zones where artificial recharge might be applied to diminish the negative impacts caused by earthquakes, it is a promising line for future research.

### 3.1.3. To Replenish Aquifers and Increase Water Availability

MAR projects in Mexico have been primarily designed to replenish aquifers and increase water availability using two types of water: stormwater and reclaimed water. Stormwater is used in small-scale facilities, mostly in rural environments, whereas in cities reclaimed water is used. In states located in the northern region, such as Baja California Sur, Chihuahua, and Sonora characterized by having a short but intense rainy season and a long dry season, there are proposals to build small-scale infrastructure to collect stormwater for replenishing aquifers [34,51,68,69]. In Chihuahua, a state located in northern Mexico, researchers from the Autonomous University of Chihuahua developed a pilot project in collaboration with the local government. This project has been ongoing for ten years [69]. In Caborca, Sonora, Minjarez et al. [68] conducted studies to develop a numerical groundwater model using stormwater as a source for MAR. Based on the model, they proposed sites for future MAR implementation. This proposal has not been transformed into a pilot project. In Southern Mexico, in a semi-arid region of the State of Oaxaca, there is a pilot project that uses stormwater for recharge [70].

In Mexico, in cities in which the population ranges from less than a million inhabitants to several million, reclaimed water is the primary source for MAR [15,37,63,71–75]. Based on the results registered by water managers, MAR using reclaimed water is more reliable because the water flow is constant, and its performance could be monitored as a component of a wastewater treatment plant.

In Mexico City, the Water System Agency has pursued a long process to create facilities for MAR using reclaimed water [63]. At the beginning of the 1960s, injection wells were used to infiltrate water from a reservoir. Even though their performance was adequate for more than nine years, these wells were closed because the water in the reservoir became contaminated [34]. In the late 1980s and early 1990s, studies were conducted to identify recharge areas in several areas of the city and the potential impacts of recharging reclaimed water [63,76]. In the eastern side of Mexico City, a suitable site for water recharge was identified. Therefore, a pilot project for MAR was built; it was running from 1992 to 2000 [63]. This installation did not operate for five years, and it was re-opened in 2005. Since 2010, these facilities located in the most populated area of Mexico City recharge reclaimed water using injection wells. This facility includes a program to monitor that the recharged water complies with the national water quality standards [63].

**Table 1.** MAR (Managed Aquifer Recharge) efforts in Mexico.

| Author/Year/State in Mexico | Objective | Specific MAR Method * | Water Source |
|---|---|---|---|
| **Research projects** | | | |
| Figueroa-Vega [34] Estado de Mexico | Reduce land subsidence and replenish the aquifer | Dug well/shaft/pit injection | Stormwater |
| González & Juárez [77] Estado de Mexico | Replenish aquifer and increase water availability | Dug well/shaft/pit injection | Stormwater |
| Minjarez et al. [68] Sonora | Replenish aquifer | Dug well/shaft/pit injection | Stormwater |
| Palma et al. [74] Hermosillo, Sonora | Replenish aquifer and increase water availability | Infiltration ponds and basins | Reclaimed water |
| Wurl [51] Wurl and Imaz-La Madrid [52] Baja California Sur | Replenish aquifer | Recharge dams | Stormwater |
| **Pilot projects** | | | |
| Gutiérrez-Ojeda [48], Gutiérrez-Ojeda and Ortiz [49] Coahuila | Restore contaminated aquifers | Infiltration ponds and dug well/shaft/pit injection | Water from a dam |
| Morales-Escalante [73] Baja California | Replenish aquifer and increase water availability | Dug well/shaft/pit injection | Reclaimed water |
| Ojeda-Olivares et al. [70] Oaxaca, Oaxaca | Replenish aquifer and increase water availability | Subsurface dams | Stormwater |
| Palma et al. [37] Chihuahua, Chihuahua ** | Replenish aquifer and increase water availability | Infiltration ponds and basins | Reclaimed water |
| Silva-Hidalgo et al. [69] Chihuahua, Chihuahua ** | Replenish aquifer and increase water availability | Subsurface dams and dug well/shaft/pit injection | Stormwater |
| **Facilities** | | | |
| Álvarez, M. [78] Querétaro | Replenish aquifer | Subsurface dams | Stormwater |
| Ávila et al. [63] Mexico City | Replenish aquifer, reduce land subsidence and increase water availability | Dug well/shaft/pit injection | Effluent |
| Briseño-Ruíz et al. [79] San Luis Potosí | Manage floods | Infiltration ponds and basins | Stormwater |
| Escolero et al. [80] San Luis Potosí | Manage floods and replenish aquifer | Infiltration ponds and dug well/shaft/pit injection | Stormwater |
| Hernández-Aguilar [36] Sonora ** | Replenish aquifer and increase water availability | Infiltration ponds and basins | Reclaimed water |
| Korenfeld & Hernández [15] | Replenish aquifer | Dug well/shaft/pit injection | Reclaimed water |
| Mendoza-Cázares et al. [64] Mexico City | Reduce land subsidence and increase water availability for environmental services | Recharge dams | Stormwater |

* International Groundwater Resources Assessment Centre-MAR classification ** Facilities/projects functioning.

The Valle de Toluca aquifer in Estado de Mexico (which is the state bordering Mexico City) is exporting water to Mexico City, even though this aquifer is experiencing overdraft. In order to increase recharge in the aquifer and reduce water stress in this basin, the government of the Estado de Mexico built facilities for MAR using reclaimed water [15]. The effluent is treated, including ultraviolet radiation to reach potable water quality standards and then injected into the ground using injection wells [15,71]. The capacity of this facility is 0.63 million cubic meters per year.

In 2007, in San Luis Río Colorado (SLRC), Sonora MAR facilities were built (Figures 2 and 3). The method used for recharge is infiltration ponds and the source of water is reclaimed water [36,72]. Annually 8.2 million cubic meters are being recharged [36]. The SLRC experience is the most successful example of MAR facilities in Mexico, using reclaimed water with the municipal government managing it.

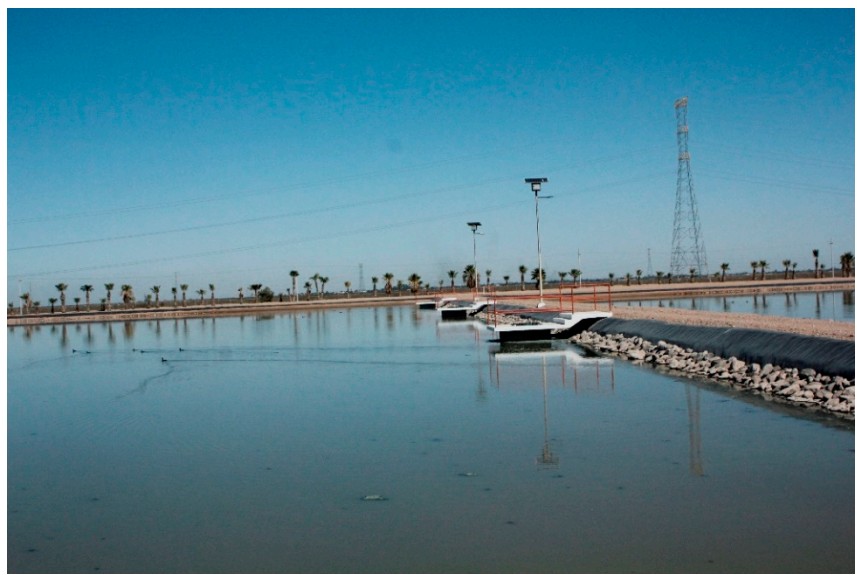

**Figure 2.** Infiltration basins in San Luis Rio Colorado, Sonora.

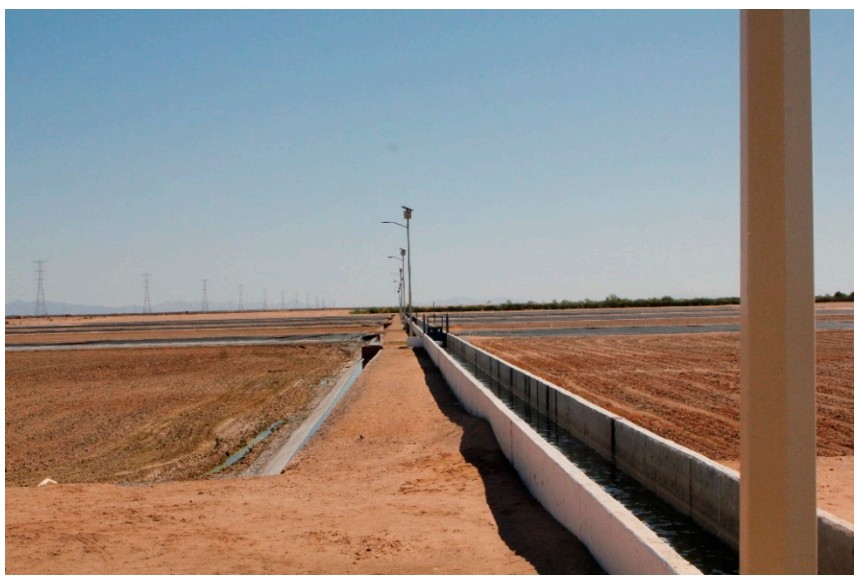

**Figure 3.** Infiltration basins in San Luis Rio Colorado, Sonora, when empty.

In the city of Chihuahua, Palma et al. [37] propose a model to recharge the unallocated effluent (about 30 million cubic meters) that is currently being discharged into the river. The water recharged would alleviate water stress in the aquifer and increase water availability for the city. In Hermosillo,

Sonora, Palma et al. [74] evaluate different scenarios to replenish the aquifer and balance the growing demand for the city's water supply. The proposal includes modifying the recharge in the irrigation districts and building MAR facilities to reduce both water loss and pollution in the system, providing additional water supply to urban areas. Based on their models, Palma et al. [74] identify the best option would be to reduce the effluent for irrigation and transfer this water to MAR facilities. This scenario means a reduction in the irrigated area, cultivating with less water, and growing a single crop each year. However, this plan would require a robust communication program with farmers and, perhaps, financial incentives to attract their participation. Another scenario would be to share the effluent between MAR and agriculture. The recharging rate will be lower compared to the other scenarios evaluated, but the recharge will be continuous [74]. The last option is the most viable because it will maintain agriculture activities and increase the groundwater recharge.

A different water source for artificial recharge was evaluated in Los Cabos, a municipality of the State of Baja California Sur. Saval [81] conducted a study using water from the San Lázaro dam. This researcher found that dam facilities were not adequate for monitoring the water recharge rate, and water quality was not satisfactory because there were anthropogenic sources of pollution. However, Saval [81] did not propose any options to eliminate these contaminants. Although, based on the information described, pollution was originated from domestic discharges which could be treated to remove contaminants.

Although there are several benefits of MAR using reclaimed water, this type of water has also generated some concerns because this water could include pollutants that cannot be removed by the soil [28,82]. To guarantee the recharged water will not affect native groundwater, the first step is to characterize the reclaimed water and include a monitoring program in the MAR project. Moreover, water composition and residence time before water is extracted for final use warrant evaluation [27,83,84]. Furthermore, new policies might be created to match water quality with native groundwater quality and the final intended use of the recharged water. For instance, water for the mining industry or farming forage crops does not require high-quality standards.

### 3.1.4. To manage Floods

Located in a semi-arid region, San Luis Potosí, capital of the state of San Luis Potosí, has faced flooding problems and water scarcity; therefore, a master plan to reduce flooding and increase water recharge was designed. For artificial recharge, the alternatives evaluated were building dams, recharge with reclaimed water, and injection wells. Based on studies conducted previously [80], a system was designed that includes a storm drain to collect water and dispose of it in ponds to infiltrate slowly [79]. This system is using facilities that previously were dedicated to sand mining, which meant a lower cost for the project. This artificial recharge project works very well; however, it does not have a program to monitor water quality.

### 3.1.5. Unintentional Recharge

Unintentional recharge refers to water discharged into a surface body or infiltrated into an aquifer, not as part of a planned project [85,86]. The Official Mexican Standard (NOM), NOM-014-CONAGUA-2003 [42] suggests that water infiltrated can be reclaimed water or non-treated water. In Mexico, one of the largest examples of unintentional recharge is in the Mezquital Valley, in the State of Hidalgo. This valley is located 80 km north from Mexico City. Since 1896, to avoid floods and to drain the sewage, a system was built for delivering non-treated water from Mexico City into the Mezquital Valley [58,87]. According to Jiménez and Chavez [87], 60 cubic meters per second are discharged daily, while during the rainy season, the peak flows may reach 300 cubic meters per second, for a few hours. Although this water contains a high percentage of contaminants that can impact the aquifer, scholars have reported that the soils have been effective in cleaning wastewater, maintaining an aquifer of acceptable quality [87]. If the water is satisfactorily treated, a portion of this resource can

be dedicated to replenishing the aquifer [88]. Hence, this environmental problem can be transformed into a solution to increase the water supply in the region.

*3.2. Water Governance and MAR*

In Mexico, the water administration is centralized in the federal government, and its regulation is grounded in Article 27 of the Political Constitution of the United States of Mexico [89]. The governance of groundwater could be summarized as having the following components:

- Legal framework: Political Constitution of the United States of Mexico, Law of the Nation´s Waters, Official Mexican standards, Federal Duties Law.
- Managers: National Water Commission and Municipalities.
- Users: Basin Councils, Groundwater Technical Councils.

3.2.1. National Regulations: Political Constitution of the United States of Mexico, Law of the Nation's Waters, and Federal Duties Law

Article 27 of the Political Constitution of the United States of Mexico (first paragraph) establishes: "The property of all land and water within national territory is originally owned by the Nation, who has the right to transfer this ownership to particulars" [89]. The sixth paragraph defines "the dominion by the State shall be inalienable and imprescriptible, and the exploitation, use or development of those resources, be that by individuals or by corporations incorporated in accordance with Mexican laws, shall not be carried out but through concessions granted by the Federal Executive in accordance with the rules and requirements so established by the laws" [89].

In 1992, Mexico adopted the Law of the Nation's Water (LAN), which together with constitutional regulations, contains specific provisions for groundwater management and the role of the National Water Commission (CONAGUA) as the federal agency responsible for water administration [41]. Article 3 of the LAN defines the types of water entitlements: appropriation, allocation, and concession.

Appropriation: can be consumptive or non-consumptive (for example for hydropower generation).

Allocation: water rights assigned to municipal and state governments for exploitation, use, and extraction of national water to be used for public services or domestic purposes.

Concession: water title assigned to public or private enterprises and citizens for exploitation, use, and extraction of national waters.

Municipal governments, state governments, and Mexico City have allocations for exploitation, use, and extraction of national water for public services and domestic utilities. Furthermore, the LAN describes that a permit issued by CONAGUA will be required to recharge reclaimed water into aquifers (article 91). Additionally, water recharged must fulfill quality parameters established in national standards.

The Federal Duties Law (LFD) contains most federal taxes and financial incentives. It is analyzed and voted on every year as a part of the national budget. The LFD includes the charges that municipal and state governments must pay to extract water (articles 223 and 223b). These charges vary according to the region where water is extracted. There are four zones, based on water availability, charges are higher where water availability is lower and vice versa. Furthermore, the LFD includes financial incentives for water recharge. Article 224, paragraph V mentions: "(Users) do not have to pay the extraction fees if the water is recharged to its original source. The water should comply with a certificate of quality recognized by CONAGUA and all physical, chemical, and biological parameters listed in Article 225". Consequently, a public or private agency that recharges water in the same aquifer, where the water was obtained, can request this payment exception or its reimbursement. In general, municipal and state governments pay water extraction fees and later demand a refund from CONAGUA equal to the fees paid.

3.2.2. Official Mexican Standards (NOMs)

NOMs are specialized regulations which contain technical details for products, processes, and services. However, they are not legislative ordinances because they were not created by both parliamentary bodies, the chamber of deputies (like the US House of Representatives) and the chamber of senators [90]. There are two NOMs concerning water quality for MAR: NOM-014-CONAGUA-2003 [42] and NOM-015-CONAGUA-2007 [43]. NOM-014 includes regulations for artificial recharge using treated wastewater. However, the NOM-014 does not include the legal definition of reclaimed water. This NOM establishes that to obtain a permit for a large-scale project, the following information is required: location, hydrogeology (piezometric and stratigraphic profile), physical and chemical characteristics of water (pH, biological oxygen demand, chemical oxygen demand, and total organic carbon) and microbiological analysis. On the other hand, NOM-015-CONAGUA-2007 describes the requirements to carry out infiltration activities using runoff to recharge groundwater. For direct aquifer recharge, the water used should meet drinking water standards.

3.2.3. Managers, CONAGUA and Municipalities

Article 4 of the LAN defines that CONAGUA as the federal agency responsible for water administration [41]. CONAGUA assumes the primary obligation to enforce laws, create a specific regulatory framework for water allocation and water quality standards that MAR projects must fulfill. Constitutional article 115 defines the responsibilities for municipal governments. Municipalities are responsible for the following functions and public services: drinking water, drainage, sewerage system, treatment, and disposal of sewage [89]. The federal government is responsible for enforcing the quality standards for drinking water. There are some participatory bodies included in the LAN, such as committees and councils that collaborate with national and state governments on water-related policies.

After examining the applicable regulations for groundwater and MAR (Table 2), we found that the LAN does not include a definition of reclaimed water and the procedures that must be followed to use it for other activities neither. There is no reference to the legal rights that public agencies have on water recharged when conducting a MAR project using wastewater and how this "new water" would be allocated. Today, municipal water agencies can use, sell, and exchange the wastewater produced in treatment plants managed by them. However, neither the CPEUM nor the LAN defines the rights and responsibilities that municipal agencies have on water recharged using a MAR method. Another omission is concerning stormwater. The LAN does not include a definition of stormwater and lacks information regarding the requirements that should be fulfilled to obtain an authorization for MAR using this type of water. Finally, we want to remark that the regulations do not establish how the stormwater recharged would be allocated or managed.

In summary, LAN lacks specific regulations for MAR. NOMs can establish directions regarding the steps to be followed for building MAR projects and define water quality standards for water recharged. However, NOMs cannot determine the requirements to obtain a permit for a MAR project or how to allocate water recharged. NOMs are complementary regulations of the laws [90], but they cannot regulate water rights.

**Table 2.** Regulations for Groundwater and MAR in Mexico, and proposals.

| | Native Groundwater | Artificial Recharge | Proposals |
|---|---|---|---|
| Definition of water rights and institutional arrangements for allocations | Concessions Assignations The Law of the Nation's Water (LAN) establishes procedures for allocations. | There is no procedure to allocate water recharged or a priority list to issue new permits. | LAN must define the procedures that must be followed to allocate water from MAR. |
| Identification of in situ requirements of available water | The National Water Commission (CONAGUA) publishes periodically water availability studies. | There is no a comprehensive study about potential water recharge projects. | CONAGUA and researchers can integrate a portfolio of viable MAR projects. |
| Abstraction limit | Entitlements are linked to a volumetric measure of water that can be extracted | The Official Mexican Standard, NOM-014-CONAGUA-2003 establishes that water recharged can be extracted after six months (surficial recharge) and 12 months of residency (direct recharge). However, there is not reference regarding the volume that can be extracted. | Based on the water rights in the basin, it could define a volume that MAR managers might extract. |
| Definition of priority uses | LAN includes a list of priority uses:<br>1. Domestic<br>2. Urban<br>3. Livestock<br>4. Agriculture<br>5. Aquaculture<br>6. Environment<br>7. Energy production for public service<br>8. Industry<br>9. Energy production for private service | None | The priority list set in the LAN might be used to define water rights to recover water recharged when using a MAR method. |
| Mechanisms for monitoring and enforcement | LAN and other federal regulations define what are the conditions under concessions can be cancelled. | LAN doesn't define penalties or mechanisms for enforcement for MAR facilities. | LAN must include mechanisms for monitoring MAR facilities. |

## 4. Discussion

In Mexico, there are several MAR pilot projects with favorable results [35]. However, only a few have been transformed into large scale facilities. In general, MAR projects have been created in arid and semiarid regions of Mexico, which have a short but intense rainy season and a long dry season. On the other hand, there are facilities in Mexico City, which is located in a region with an oceanic climate (Cwb). This city registers precipitation throughout the entire year [91] and on average is 600 mm per year.

We found examples of MAR projects in 10 states; these projects have primarily been designed to recover aquifers and to prevent floods [70], and local water agencies are responsible for them [25,34,68,69]. The experience and knowledge that local authorities have acquired are valuable resources and might be integrated by the federal government when designing MAR policies. Local agencies know how to create these types of facilities and are aware of the challenges they face in maintaining them.

Three MAR facilities using reclaimed water were operating. These facilities are located in Toluca, Estado de Mexico, Iztapalapa, Mexico City, and San Luis Río Colorado (SLRC), Sonora. To date, only San Luis Rio Colorado MAR facility is functioning. Another example is a pilot project located in the city of Chihuahua that uses reclaimed water [37,92]. Several cities in the United States of America use effluent as an additional source for water supply [31–33]. For instance, in Tucson, Arizona, Tucson Water (the principal water utility in the city) has integrated reclaimed water as a source for future water supply [32,93,94].

One of the largest facilities for MAR in the world, that uses reclaimed water, is located in Orange County, California. The effluent is treated to reach potable water quality standards and then injected into the aquifer [33]. MAR is helping to recover the aquifer and diminish saline intrusion [95]. In the northwest region of Mexico and other states located in the Yucatan Peninsula, several coastal aquifers are affected by seawater intrusion [1,8]. Cardona et al. [50] and Wurl [51] have studied seawater intrusion in Baja California Sur (located in the northwest region) and proposed that MAR can be a tool to address this problem. It is a promising research field for MAR in Mexico that might be explored in the future.

### 4.1. MAR in the Metro Area of Mexico City

Mexico City and its metropolitan area, the so-called Metropolitan Zone of the Valley of Mexico (MZVM), faces floods almost every summer because there is no infrastructure to manage and store stormwater. Since the early 1940s, the MZVM began importing small volumes of water from the Estado de Mexico [1]. This decision was made because of the rapid increase in the population that demanded more water and the subsidence problems caused by overexploitation of groundwater from the local aquifer [1,66,76]. To the present, more than 25 percent of the total volume used for human consumption and productive activities is imported. About 65 percent of the water used is groundwater pumped from the local aquifer [1]. At the MZVM, several researchers and consultants have carried out experiments and designed MAR pilot projects to reduce land subsidence [34,63,76]. Given that the rate of subsidence is high, small-scale infrastructure could be insufficient to solve this problem in the near future. However, even little steps can help to reduce the subsidence problem.

For more than 100 years, in Mexico City and the MZVM, the stormwater reclaimed water and untreated water have been blended to be discharged into streams outside of the area, producing positive and negative impacts [1,59,76,88]. On the one hand, the aquifer has been recharged, and the water table has increased. On the other hand, the soil has been contaminated because a portion of the water is not adequately treated. Jiménez [88] found that the water in the aquifer has better quality compared to the inflow; it means that the soil has functioned as a cleaning system. To date, farmers are entitled to a large percentage of this water. Jiménez [88] mentioned that significant volumes of water are wasted because there is no infrastructure for irrigation. If the water is satisfactorily treated, a portion of this resource can be dedicated for MAR [88]. In the future, this scenario would be

possible because a new facility was built and, now, around 60 percent of the water from the MZVM is being treated [96,97]. The new wastewater treatment plant is designed to treat 35,000 L per second. Considering that the MZVM lacks water, part of the reclaimed water can be delivered to agricultural activities, and another percentage might be utilized for MAR. It would mean an additional source of water for this thirsty metropolis.

### 4.2. MAR Using Stormwater

Generally, it has been suggested that, before starting a MAR project using stormwater, the following actions must be conducted: characterize the water, create an inflow map, and carry out pre-treatment procedures [98]. These steps are needed because stormwater, from urban areas, may contain contaminants such as oil, grease, metals, and pesticides [98,99]. Globally, there are successful examples of MAR using stormwater from urban environments. For instance, in Santa Cruz County, California, a pilot project composed of a network of basins for MAR has shown positive results. This program is managed by the Resource Conservation District of Santa Cruz County, and it is designed to recover groundwater and collect excess runoff for mitigating flooding as well [100,101]. For the city of Rome, Italy, La Vigna et al. [102] proposed building infrastructure for MAR using stormwater as source for recharge. In Adelaide, Australia MAR facilities have been constructed using stormwater from peri-urban regions [103]. In Mexico, we found that the infrastructure for MAR using stormwater is composed of small-scale installations constructed by local communities, non-profit organizations, and researchers [64,69,70]. Non-centralized systems like these examples offer several benefits relative to a centralized MAR system because these installations take advantage of natural precipitation and flow pathways, they can be developed and operated at relatively low cost [100].

In the cities of Chihuahua, Oaxaca, Mexico City (rural area), and San Luis Potosi, there are pilot projects and small-scale facilities that use stormwater to recharge aquifers [52,64,70,79]. MAR infrastructure has been constructed outside the urban grid or in rural areas, except in the case of San Luis Potosí, where the objective is to reduce floods in the city. Therefore, it is essential to acknowledge the role that these projects have to increase water availability, alleviate water scarcity, provide environmental services in riparian ecosystems, and improve water use in agriculture.

Regarding the role that reclaimed water can play to provide environmental services, there are limited examples. In Tucson, Arizona, there is an effluent-dependent riparian ecosystem that provides environmental services and generates social benefits such as recreation [104,105]. In Mexico City, in a rural community within this city, researchers from the Mexican Institute for Water Technology built infrastructure to recharge water [64]. Although this recharge is not directly increasing the water availability for human consumption, in the long-term, this water will help to recover the local aquifer. Besides, the recharging process by itself is providing water to maintain environmental services.

Soft-path solutions for water management include the creation of small-scale sources for supply, methods to increase efficiency to meet water demands, and the design of new policies [106,107]. In Mexico, it must be acknowledged that the small-scale facilities can be part of the portfolio for better groundwater management. These facilities are part of the soft-path solutions that can be better integrated into national or regional policies to increase MAR projects.

### 4.3. Legal Framework and Public Policies for MAR

Worldwide, scientific knowledge in hydrology and geology has expanded, but the governing institutions responsible for making decisions about groundwater have been slowly improved [108,109]. Moreover, the Organisation for Economic Cooperation and Development (OECD) has expressed that "the current water crisis is not a crisis of scarcity but a crisis of mismanagement, with strong public governance features" [110–112]. It has been remarked that adequate policies and guidelines must be in place to ensure the benefits that MAR can provide [113]. In Mexico, there are more than 50 years of technical experience in MAR research [35,51]. Nevertheless, the information generated during this

period has not been sufficiently incorporated into the national policy for MAR to launch long-term projects or propose public policies to increase its practice.

Mexican researchers have provided evidence to support MAR as a feasible option to increase water supply [114]. Recharging facilities are helping in increasing water availability in arid regions, such as the San Luis Rio Colorado, Sonora, and Chihuahua examples [36,69]. MAR facilities in Mexico City are aiding to diminishing the adverse effects caused by floods and reducing the rate of subsidence [65,80]. Two of the most successful MAR projects utilize reclaimed water, and there are still other MAR proposals that could be implemented [37,73]. However, the regulatory framework needs clarification and improvement to succeed in implementing MAR projects.

In Mexico, all the waters are owned by the nation; therefore, to be used or exploited the LAN includes three types of water entitlements: appropriation, allocation, and concessions [41]. In general, municipal operators have appropriations which cannot be transferred or changed to other uses than those for human provision (LAN). Since the Mexican legal framework lacks definitions for entitlements to recover stored groundwater using MAR methods, some options to protect investment for MAR projects must be evaluated. The security of recovery entitlements for MAR operators is an essential consideration for investment [115].

The Law of the Nation's Waters does not include a definition for reclaimed water and lacks procedures to define how reclaimed water can be managed and allocated. Gilabert-Alarcon et al. [116] highlight that policies and regulations for reclaimed water are limited. Considering that reclaimed water can be used for agricultural activities and MAR projects, the lack of a regulatory framework is hampering the opportunities for reusing this water. The responsible use of reclaimed water would generate economic and environmental benefits such as reducing water stress and preventing seawater intrusion in coastal aquifers [116]. In Mexico, the absence of a legal definition for reclaimed water and how it can be allocated generates uncertainty regarding water rights. This gap must be solved when creating national policies for MAR.

Another absence in the legal framework is that neither the Political Constitution of the United States of Mexico or the Law of the Nation's Water (LAN) defines the rights and responsibilities that municipal agencies have on water recharged, using a MAR method. Usually, municipal and state water agencies responsible for wastewater treatment plants exchange reclaimed water with farmers. Furthermore, these agencies sell reclaimed water to industries, golf courses, and construction companies. Nevertheless, if wastewater is recharged, municipal or state agencies lose their rights. This situation is the same if stormwater is used. A water right is a legal entitlement tied to a user and to a volume allocated by CONAGUA in each basin. Only the National Water Commission (CONAGUA) could define how the recharged water can be apportioned. However, Mexican legislation does not establish the guidelines that CONAGUA should follow in this regard. This lack of definition is limiting the participation of municipal water agencies in MAR projects.

Nowadays, it is recognized that scientific and technical information is not enough to address the complexity of groundwater management. It is vital to collaborate with society to solve real-world problems [117]. In Mexico, a single agency, CONAGUA, defines groundwater policies and guidelines for their implementation. There is a single policy for the whole country. The national government has a central vision for groundwater management and MAR that applies to the entire country, regardless of local conditions. An effective groundwater governance framework acknowledges and incorporates the local and regional socio-cultural values of water [118]. Groundwater resources in Mexico are facing tremendous pressure; 70 percent of the water used for agriculture and human consumption is groundwater [1]. It would be desirable that CONAGUA initiates a dialogue with local governments, users, and researchers to design MAR rules. The success or failure of natural resources management initiatives promoted by central authorities strongly depends on the support provided to local governments [119].

Regarding public policies for MAR and decentralization in Mexico, the challenge is how to incorporate in the decision-making process, the knowledge acquired by local governments,

without amending the legal framework for groundwater. As Nagendra & Ostrom [119] described in forest management, the interactions between actors from several agencies such as federal, state, and municipal could be more important than the changes in the formal legal structure. In Mexico, it is not required to amend the legislation and decentralize responsibilities to increase the participation of both local government and citizens in strengthening public policies for MAR.

Mexican scholars investigating groundwater and MAR recognize that a multidisciplinary approach needs to be taken regarding groundwater management and MAR policies [22,120]. Furthermore, scholars suggest that financial incentives and new legislative rules for stimulating the creation of MAR facilities are required [34,69,120]. However, academics do not specify how the legislation might be improved or the type of incentives that are needed. Allegedly, the current legal framework in Mexico is limiting the creation of MAR projects. Therefore, beyond technical experience or knowledge, today, there is a need to evaluate and update the national legal framework to incorporate artificial recharge methods as part of the solutions to increase water availability.

*4.4. Legal Framework That Might Be Revised and Proposals*

Allocation of the recharged water. Inadequate institutional arrangements for aquifer storage and extraction can lead to a legal dispute and potentially negatively impact natural environments [115,121]. Allocation of water recharged is an issue that has been addressed using different tools. In the US, for instance, the states of Arizona, California, and Florida [113,122–124] have created a complex legal framework that establishes the permits required to recharge and extract water for MAR projects.

In Mexico, the Law of the Nation's Water (LAN) establishes that groundwater might be assigned based on the availability calculated by CONAGUA [41]. If there is unallocated water, it might be apportioned according to the priority list established in the LAN. The use of groundwater for domestic and urban use has primacy over other uses. However, the regulatory framework does not define how to allocate water that has been recharged using artificial methods.

The priority list set in the LAN would be used to determine water rights for recovering water recharged when using a MAR method. It would be possible to consider incorporating in the Mexican legal framework a general concept proposed by Pyne [125] that "if a water user has a right to the water before water recharge, then the user also has a right to recover that water". It would mean that the recharged water would be assigned using the same priority list that the LAN establishes when it is extracted from a natural source. This regulation would help to protect investors that are recharging water.

The institutional arrangement for MAR in Mexico is in its early stages, while federal regulations include some ordinances for recharge projects, there are no provisions for the recovery of stored water. In addition, considering that all waters are public property, there is limited space to create legal tools to protect water volumes for users who lack concessions or assignations. Under the current legal framework, an option that could be explored is allowing that those water users, entitled with allocations and that are recharging water, to extract a specific volume in addition to their allocation. In South Australia, for example, in some MAR projects carried out in overexploited aquifers, a recovery rate of 80% has been established [115]. In Mexico, the percentage of water that could be extracted would depend on the specific conditions of the aquifer and the type of water right (domestic, agriculture, industry).

Demonstration projects. One option to promote MAR methods could be to use successful experiences as demonstration projects. There is an example of this type of public policy in the United States (US). In 1984, the US Congress enacted the High Plains States Groundwater Demonstration Program Act. Federal agencies conducted studies in 17 states to identify appropriate sites for MAR. A report developed by Rogers [126] pointed out that "aquifer recharge programs are a good example of federal-local partnership that result in long-term investments."

Another example is Arizona because, as part of the legislation enacted in the 1990s, specific provisions for state demonstration projects were included [124,127]. Based on this legislation, MAR demonstration projects were built to show their effectiveness and technical aspects to operate them.

Arizona's demonstration project program incorporated both facilities permitting and operation provisions. Furthermore, in Arizona, the State Demonstration Recharge Program was funded through the Central Arizona Water Conservation District [122,124,127,128]. In the state of California [129,130], the legislation includes demonstration MAR projects. Furthermore, the Australian Guidelines for MAR include a commissioning phase for MAR projects as a tool to manage risk for projects in the development stage [86].

In Mexico, it could be feasible to incorporate demonstration projects into national policies. The infrastructure will be publicly owned; such has been proposed in Australia for stormwater harvesting [121]. Local demonstration projects that present technical and cost information are essential to draw the attention of a broad audience and increase the chance that MAR can be adopted [26]. Additionally, these facilities would help to create a link between MAR researchers and water managers. In summary, it would be desirable to include specific provisions in the legislation or create public programs to promote successful experiences, show their effectiveness, and technical aspects to operating them in Mexico.

## 5. Conclusions

Undoubtedly, one way to ensure water supply in the future is by reducing groundwater demand [23]. For instance, increasing the efficiency in agriculture and recycling water in industrial activities where it is feasible. Furthermore, programs to improve water replenishment, such as MAR projects and new public policies, are needed. In Mexico, previous research has been focused on assessing the performance of MAR projects or suitability studies. To the best of our knowledge, this paper represents the first academic effort to evaluate the role that governance plays in augmenting water supply sources in arid and semi-arid regions in Mexico, using MAR methods.

In Mexico, since the 1950s, researchers, consultants, and local governments have designed and implemented water recharge projects [25]. Even though some cases were not designed specifically for water recharge, it has been pointed out that these experiences revealed the potential of collecting and storing water to be used in the future [34]. We found that beyond the technical issues that MAR projects normally address, the regulatory framework is a barrier to increasing MAR facilities because there are no provisions for the recovery of stored water.

Two of the most successful MAR projects, in terms of the amount of water recharged, use reclaimed water, and local water agencies are responsible for these facilities. Recharging facilities are helping in recovering water-deficit in arid regions in Mexico, such as the San Luis Rio Colorado, Sonora, and Chihuahua cases. However, the Law of the Nation´s Waters does not include a definition for reclaimed water and lacks procedures to define how reclaimed water can be managed and allocated. Gilabert-Alarcon et al. [116] highlight that policies and regulations for reclaimed water are limited. Considering that reclaimed water can be used for agricultural activities and MAR projects, the lack of a regulatory framework is hampering the opportunities for reusing this water.

MAR facilities in Mexico City using stormwater have been valuable tools for diminishing the adverse effects caused by floods and helping to reduce the subsidence rate [25,35]. Local governments are responsible for these facilities. Hence, the experience of municipal water agencies should be incorporated when designing policies for MAR.

We found several cases in rural communities and in medium-sized cities where small-scale projects for MAR have been built. These facilities have been created to recover aquifers, and local authorities managed them. It is essential to recognize the role of small-scale projects, as part of soft-path solutions, can play to increase water supply [131]; also, how these installations can be incorporated into regional water portfolios.

It is critical for water-stressed countries like Mexico to contemplate MAR as an option to increase water availability. It has been remarked that adequate policies and guidelines must be in place to ensure the benefits that MAR can provide [113]. An effective groundwater entitlement scheme is

required to ensure investment in MAR projects [115]. In Mexico, beyond engineering knowledge regarding MAR, specific federal policies addressing gaps in the legislation are needed.

**Author Contributions:** This study was designed by M.B.C.-A., under the supervision of S.B.M. Writing—Original Draft Preparation, M.B.C.-A.; Review & Editing S.B.M. All authors have read and agreed to the published version of the manuscript.

**Funding:** Partial funding for this research was provided by the Inter-American Institute for Global Change Research CRN3056 Project (supported by NSF Grant No. GEO-1128040) via support from the Udall Center for Studies in Public Policy and the Tinker Field Research Grant (Summer 2019) via the Center for Latin American Studies at the University of Arizona.

**Acknowledgments:** M.B.C.A. thanks the Mexican Council for Science and Technology (CONACYT) for the fellowship to M.B.C.A. through a Ph.D. grant, the Graduate & Professional Student Council at the University of Arizona and The Herbert E. Carter Travel Award Program for providing partial funding to present this research at the ISMAR 10 conference. We want to thank M. Wilder for her helpful recommendations and the three anonymous reviewers who helped to improve this paper with their accurate and insightful suggestions.

**Conflicts of Interest:** The authors declare no conflict of interest. The founding sponsors had no role in the design of the study; in the collection, analysis, or interpretation of data; in the writing of the manuscript, and in the decisions to publish the results.

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
