# Peer review of "An Overview of Managed Aquifer Recharge in Mexico and Its Legal Framework"

_water, doi:10.3390/w12020474_

Round 1
Reviewer 1 Report
This paper fils a very useful gap in terms of global MAR awareness and its challenges to implementation. Most of my comments are minor in nature. However, some issues arise when using projects within the United States as regulatory examples for MAR. This is because in the US, ASR (injection wells) is managed differently due to federal requirements (Class IV) - because of this, some states do not regulate "MAR" - they regulate certain types of MAR. This may need to be noted or explained a bit more in this paper.
Line 164: Why did this project not move forward? That detail would be helpful to your readers.
Line 172. "It because" needs revising
Line 176: Insert "MAR to reduce land subsidence"
Line 183: Due "to" is missing
Line 218 through 222: Those 2 sentence needs to be revised - a sentence within a sentence is hard to understand and read.
Line 226: remove "water on the aquifer". It is not necessary and 'water on" an aquifer is not an accurate description of the hydrology of MAR.
Line 448 through 450. The sentence, "For instance..." Is this a rural community or a community within Mexico City? Was infrastructure built for MAR by this community? This sentence requires revision.
Line 549. Need to add references to the end of the "..replenish aquifers" sentence. As no mention of Washington was previously discussed. Also in Washington, the state regulates only injection wells (ASR), shallow aquifer recharge does not fall under state permitting requirements. The groundwater reservoir permit and respective legislation do provide a somewhat streamline process to recharge aquifers, but again, this is only for ASR - not MAR.
Line 560: "Besides of that" needs to be revised.
Author Response
Dear reviewer,
We are grateful for all your insightful comments and suggestions, which have helped us to improve the manuscript significantly. We have carefully taken your feedback into consideration and modified the document accordingly (please see the attachment).
Kind Regards,
MB

Reviewer 2 Report
The authors provide a comprehensive review of MAR projects in Mexico and attendant legislative arrangements. The manuscript is well structured and thoughtful and makes a useful contribution to the literature concerned with MAR. The review and analysis of the Mexican legislative framework as a primary impediment to MAR development and operations is an important contribution, correcting an aspect that receives minimal attention.
Comments for author and editor consideration.
Line 54: do you mean return flows from irrigation and liquid waste disposal?
line 111: consider differentiating and defining water availability, access and water security or even water poverty. Is security, access or availability the central challenge here that MAR can mitigate?
Line 119: The "paper" does not analyze, the authors analyze and the results are reported in the paper. Stating "the paper does...." humanizes the manuscript. Suggest rewording.
Line 169: Defining what the authors mean by "recovered" is a necessary step here. eg sentence below (recovered to 18% of depletion). Best to state " the last option would result in 18% of depleted groundwater compared to x% from other MAR options.
Line 219: Better suited to a footnote
Line 271: more detail re soil percolation and contaminant reduction/removal useful here (eg see Diilon Pavelic Page inter alia.
Line 322: suggest describing the Legislative definition of entitlements, allocations, concessions or licenses. These are central to understanding groundwater governance and by extension MAR. By so doing, the authors provide a reference point for suggested improvements/defining of rights to the operational stages of MAR.
Line 482: what would be the attributes of an adequate legal framework? This is the opportunity for the authors to make a substantive contribution to the MAR literature. With a thorough background review, I argue the authors have missed the opportunity
Line 489: important point. Defining or expanding on "rights" would be useful for those technical readers not familiar with generally held interpretations.
Line 499-503: Here is an opportunity to adapt international examples of MAR institutional arrangements and rights to resolve/mitigate the identified deficiencies of Mexican MAR related legislature and regulations.
562: the claim to be the first academic effort is a bold one and likely to be contested. Much has been written on MAR in countries typically defined as arid and semi arid (Australia, USA, India, Pakistan. Africa). Perhaps arid and semi-arid regions in Mexico wold be more appropriate.
These may be useful (also Shah and Maheshwari et al. for Indian examples
1. Dillon, P. D Page, J Vanderzalm, S Toze, J Ward (2010) The pathway from boutique to mainstream water supply. AWA Journal of Water: April 2010.
2. Ward, J and Dillon (2012) Design principles to coordinate managed aquifer recharge with natural resource management policies in Australia. Hydrogeology Journal 20 (5) 943-956
3. Ward, J, Dillon, P. and Grandgirard, A. (2008) Designing tradeable rights to manage aquifer recharge according to robust separation principles, Water, Science and Technology: Water Supply 8(4).
Author Response
Dear Reviewer,
Thank you for reading our manuscript and reviewing it. We have considered all your comments carefully and responded appropriately. We appreciate your valuable suggestions to improve the paper.
Sincerely,
MBCA and SBM

Reviewer 3 Report
Please report and make reference about IGRAC MAR classification.
This work is similar to:
González Villarreal, F., Cruickshank Villanueva, C., Palma Nava A., Mendoza Mata, A. (2015). Recarga Artificial de Acuíferos en México. Revista H2O, del Sistema de Aguas de la Ciudad de México. Año 2. Enero-Marzo.
Please review it.
Author Response
Dear reviewer,
We want to thank you for reading our manuscript and reviewing it. We have incorporated your suggestions and added the reference suggested.
Respectfully,
Here are the responses to your comments and suggestion:
1. Please report and make a reference about the IGRAC MAR classification.
Response 1: We have included the IGRAC-MAR classification (Table 1), lines 259-260
2. This work is similar to:
González Villarreal, F., Cruickshank Villanueva, C., Palma Nava A., Mendoza Mata, A. (2015). Recarga Artificial de Acuíferos en México. Revista H2O, del Sistema de Aguas de la Ciudad de México. Año 2. Enero-Marzo [reference 35].
Response 2: many thanks for your suggestion; we have included this reference in the manuscript.
Round 2
Reviewer 3 Report
No suggestions